# TRAINING COMPRESSED FULLY-CONNECTED NETWORKS WITH A DENSITY-DIVERSITY PENALTY

**Shengjie Wang**
Department of CSE
University of Washington
wangsj@cs.washington.edu

**Haoran Cai**
Department of Statistics
University of Washington
haoran@uw.edu

**Jeff Bilmes**
Department of EE, CSE
University of Washington
bilmes@uw.edu

**William Noble**
Department of GS, CSE
University of Washington
william-noble@u.washington.edu

## ABSTRACT

Deep models have achieved great success on a variety of challenging tasks. However, the models that achieve great performance often have an enormous number of parameters, leading to correspondingly great demands on both computational and memory resources, especially for fully-connected layers. In this work, we propose a new "density-diversity penalty" regularizer that can be applied to fully-connected layers of neural networks during training. We show that using this regularizer results in significantly fewer parameters (i.e., high sparsity), and also significantly fewer distinct values (i.e., low diversity), so that the trained weight matrices can be highly compressed without any appreciable loss in performance. The resulting trained models can hence reside on computational platforms (e.g., portables, Internet-of-Things devices) where it otherwise would be prohibitive.

## 1 INTRODUCTION

Deep neural networks have achieved great success on a variety of challenging data science tasks (Krizhevsky et al., 2012; Hinton et al., 2012; Simonyan & Zisserman, 2014b; Bahdanau et al., 2014; Mnih et al., 2015; Silver et al., 2016; Sutskever et al., 2014). However, the models that have achieved this success have a very large number of parameters, a consequence of their wide and deep architectures. Although such models yield great performance benefits, the corresponding memory and computational costs are high, making such models inaccessible to lightweight architectures (e.g., portable devices, Internet-of-Things devices, etc.). In such settings, the deployment of neural networks offers tremendous potential to produce novel applications, yet the modern top-performing networks are often infeasible on these platforms.

Fully-connected layers and convolutional layers are the two most commonly used neural network structures. While networks that consist of convolutional layers, are particularly good for vision tasks, the fully-connected layers, even if they are in the minority, are responsible for the majority of the parameters. For example, the VGG-16 network (Simonyan & Zisserman, 2014a) has 13 convolutional layers and 3 fully-connected layers, but the parameters for 13 convolutional layers contain only $\sim 1/9$ of the parameters of the three fully-connected layers. Moreover, for general tasks, convolutional layers may be not applicable (data might be one-dimensional, or there might be no local correlation among data dimensions). Therefore, compression of fully-connected layers is critical for reducing the memory and computational cost of neural networks in general.

We can use the characteristics of convolutional layers to reduce the memory and computational cost of fully-connected layers. Convolution is a special form of matrix multiplication, where the weights of the matrix are shared according to the convolution structure (low diversity), and most entries of the weight matrix are zeros (high sparsity). Both of these properties greatly reduce the information capacity of the weight matrix.

In this paper, we propose a density-diversity penalty regularization, which encourages low diversity and high sparsity on fully-connected layers. The method uses a pairwise L1 loss of weight matrices to the training objective. Moreover, we propose a novel "sorting trick" to efficiently optimize the density-diversity penalty, which would otherwise would be very difficult to optimize and would slow down training significantly. When initializing the weights to have a small portion of them to be zero, the density-diversity penalty effectively increases the sparsity of the trained weight matrices in addition to reducing the diversity, generating highly compressible weight matrices.

On two separate tasks, computer vision and speech recognition, we demonstrate that the proposed density-diversity penalty significantly reduces the diversity and increases the sparsity of the models, while keeping the performance almost unchanged.

## 2 PREVIOUS RELATED WORK

For this paper, we focus on reducing the number of actual parameters in fully-connected layers by using the density-diversity penalty to encourage sparsity and penalize diversity. Several previous studies have focused on the task of reducing the complexity of deep neural network weights.

Nowlan & Hinton (1992) introduced a regularization method to enforce weight sharing by modeling the distribution of weight values as a Gaussian mixture model. This approach is in the same spirit as our method, because both methods apply regularization during training to reduce the complexity of weights. However, the Nowlan et al. approach focuses more on the generalization of the network rather than the compression. Moreover, their approach involves explicitly clustering weights to group them, while in our approach, weights are grouped by the regularizer directly.

The "optimal brain damage" (LeCun et al., 1989) and "optimal brain surgeon" (Hassibi & Stork, 1993) methods are two approaches for pruning weight connections of neural networks based on information about the second order derivatives. Interestingly, Cireşan et al. (2011) showed that dropping weights randomly can lead to better performance. All three approaches focus more on pruning unnecessary weights of the network (increasing sparsity) rather than grouping parameters (decreasing diversity), while our approach addresses both tasks with a single regularizer.

Chen et al. (2015b) proposed a hashing trick to randomly group connection weights into hash buckets, so that the weight entries in the same hash bucket are tied to be a single parameter value. Such approaches force low diversity among weight matrix entries, and the grouping of weight entries are determined by the hash functions prior to training. In contrast, our method learns to tie the parameters during training.

Han et al. (2015b) first proposed a method to learn both weights and connections for neural networks by iteratively pruning out low-valued entries in the weight matrices. Based on that idea, they proposed a three stage pipeline (Han et al., 2015a): pruning low-valued weights, quantizing weights through $k$-means clustering, and Huffman coding. The resulting networks are highly compressed due to both sparsity (pruning) and diversity (quantization). The quantization of weights is applied to well-trained models with pruned weights, whereas our method prunes and quantizes weights simultaneously during training. Our approach is most relevant to that of Han et al., and we achieve comparable or better results. In some sense, it may be argued that our approach generalizes on that of Han et al. because the hyperparameter controlling the strength of the density-diversity penalty can be adaptively increased during training (although in the present paper, we keep it fixed during training).

## 3 DENSITY-DIVERSITY PENALTY

Suppose we are given a deep neural network of the following form:

$$\hat{y} = W_m \phi(W_{m-1} \phi(\ldots \phi(W_1 x))), \qquad (1)$$

where $W_j$ represents the weight matrix of layer $j$, $\phi$ is a non-linear transfer function, $x$ denotes the input data, and $y$ and $\hat{y}$ denote the true and estimated labels for $x$, respectively. Suppose weight matrix $W_j$ has order $(r_j, c_j)$.

Let the objective of the deep neural network be $\min L(\hat{y}, y)$, where $L(\cdot)$ is the loss function. We propose the following optimization to encourage low density and low diversity in weight matrices:

$$\min L(\hat{y}, y) + \sum_{j=1}^{m} \lambda_j \left( \sum_{\substack{a=1:r_j, \\ b=1:c_j}} \sum_{\substack{a'=1:r_j, \\ b'=1:c_j}} |W_j(a,b) - W_j(a',b')| + \|W_j\|_p \right), \qquad (2)$$

where $W_j(a,b)$ denotes the entry of $W_j$ in the $a^{\text{th}}$ row and $b^{\text{th}}$ column, $\lambda_j > 0$ is a hyperparameter, and $\|\cdot\|_p$ is a matrix $p$-norm (e.g., $p = 2$ gives the Frobenius norm, or $p = 1$ is a sparsity encouraging norm). We denote the density-diversity penalty for each $W_j$ as $DP(W_j)$.

In general, for weight matrix $W_j$, the proposed density-diversity penalty resembles the pairwise L1 difference over all entries of $W_j$. Intuitively, such regularization forces the entries of $W_j$ to collapse into the same values, not just similar values. The regularizer thus reduces the diversity of $W_j$ significantly. The diversity penalty, therefore, is a form of total variation penalty (Rudin et al., 1992) but where the pattern on which total variation is measured is not between neighboring elements (as in computer vision (Chambolle & Lions, 1997)) but rather globally among all pairs of all elements in the matrix.

Though the hyperparameter $\lambda_j$ can be tuned for each layer, in practice, we only tune one $\lambda_j$ for layer $j$, and for all layers $j' \neq j$, we set $\lambda_{j'} = (\frac{r_{j'}c_{j'}}{r_j c_j})\lambda_j$, because the magnitude of the density-diversity penalty is directly correlated with the number of entries in the weight matrix.

### 3.1 Efficiently Optimizing the Density-Diversity Penalty

While the gradient of the $p$-norm part of the density-diversity penalty is easy to calculate, computing the gradient of the diversity part of the density-diversity penalty is expensive: naive evaluation costs $O(r_j^2 c_j^2)$ for a weight matrix $W_j$ of order $(r_j, c_j)$. If we suppose $r_j = 2000$ and $c_j = 2000$, which is common for modern deep neural networks, then the computational cost of the density-diversity penalty becomes roughly $(2000)^4 = 1.6 \times 10^{13}$, which is intractable even on modern GPUs.

For simplicity, suppose we assign the subgradient of each L1 term at zero to be zero, which is typical in certain widely-used neural network toolkits such as Theano (Theano Development Team, 2016) and mxnet (Chen et al., 2015a). With a sorting trick, shown in Alg 1, we can greatly reduce the computational cost of calculating the gradients of density-diversity penalty from $O(r_j^2 c_j^2)$ down to only $O(r_j c_j (\log r_j + \log c_j))$.

> **input** : $W_j, \lambda_j, r_j, c_j$
> **output**: $\frac{\partial DP(W_j)}{\partial W_j}$
> $\bar{W}_j = \text{flatten}(W_j)$ ;
> $I_j = \text{sort\_index}(\bar{W}_j)$ ;
> $I'_j = \text{num\_greater}(I_j)$;
> $\frac{\partial DP(W_j)}{\partial W_j} = \text{reshape}(\lambda_j * (I'_j - I_j), (r_j, c_j))$;
> **return** $\frac{\partial DP(W_j)}{\partial W_j}$

**Algorithm 1:** Sorting Trick for Efficiently Calculating the Gradient of Density-Diversity Penalty on Weight Matrix $W_j$ : $\frac{\partial DP(W_j)}{\partial W_j}$

In the algorithm, flatten($W_j$) transforms the matrix $W_j$, which is of order $(r_j, c_j)$, into a vector of length $r_j * c_j$, and sort\_index($\cdot$) outputs the sorted indices (in ascending order) of the input vector, e.g. sort\_index$((3, 2, 1, 2, 3, 4)) = (3, 1, 0, 1, 3, 5)$. In other words, entry $i$ in the sorted indices is the number of entries in the original vector smaller than the $i$th entry. Also note that the entries with the same value have the same sorted index. Correspondingly, num\_greater($\cdot$) outputs the number of elements greater than a certain entry based on the sorted index. For example, num\_greater$((3, 1, 0, 1, 3, 5)) = (1, 3, 5, 3, 1, 0)$. Finally, reshape($\cdot$) transforms the input vector into a matrix of the given shape.

The computational cost of the sorting trick is dominated by the sorting step $I_j = \text{sort\_index}(\bar{W}_j)$, which is of complexity $O(r_j c_j (\log r_j + \log c_j))$. We show that the sorting trick outputs the correct

gradient in the following equation:

$$
\begin{aligned}
\frac{\partial DP(W_j)}{\partial W_j(a,b)} &= \frac{\partial \sum_{a'=1:r_j, b'=1:c_j} |W_j(a,b) - W_j(a',b')|}{\partial W_j(a,b)} \\
&= \sum_{W_j(a,b) > W_j(a',b')} \frac{\partial |W_j(a,b) - W_j(a',b')|}{\partial W_j(a,b)} + \sum_{W_j(a,b) < W_j(a',b')} \frac{\partial |W_j(a,b) - W_j(a',b')|}{\partial W_j(a,b)} \\
&= \sum_{W_j(a,b) > W_j(a',b')} 1 + \sum_{W_j(a,b) < W_j(a',b')} -1 \\
&= I'_j(ar_j + b) - I_j(ar_j + b),
\end{aligned}
\tag{3}
$$

where $I_j(ar_j + b)$ corresponds to the $(ar_j + b)$th entry of $I_j$, which is the $a$th row and $b$th column of the matrix formed by reshaping $I_j$ into $(r_j, c_j)$.

Intuitively, $\frac{\partial DP(W_j)}{\partial W_j(a,b)}$ requires counting the number of entries in $W_j$ with values greater than $W_j(a,b)$, and the number of entries less than $W_j(a,b)$. By sorting entries of $W_j$, we get $I_j(ar_j + b)$ and $I'_j(ar_j + b)$ for all pairs of $(a,b)$ collectively; therefore, we can easily calculate the gradient for the density-diversity penalty.

Although the sorting trick is efficient for calculating the gradient for the density-diversity penalty, depending on the size of each weight matrix, the computational cost can still be high. In practice, to further reduce computational cost, for every mini-batch, we only apply the density-diversity penalty with a certain small probability (e.g. 1% to 5%). This approach still effectively forces the values of weight matrices to collapse, while the increase in the training time is not significant. For our implementation, to accelerate collapsing of weight entries, we truncate the weight matrix entries to have a limited number of decimal digits (e.g. 6), in which case entries with very small differences (e.g. $1e - 6$) are considered to be the same value.

## 3.2 ENCOURAGING SPARSITY

The density-diversity penalty forces entries of a weight matrix to collapse into the same value, yet sparsity is not explicitly enforced. To encourage sparsity in the weight matrices, we randomly initialize every weight matrix with 10% sparsity (i.e., 90% of weight matrix entries are non-zero). Thereafter, every time we apply density-diversity penalty, we subsequently set the weight matrix value corresponding to the modal value to be zero. Because the value zero is almost always the most frequent value in the weight matrix (owing to the $p$-norm), weights are encouraged to stay at zero when using this method, because the density-diversity penalty encourages weights to collapse into same values. Our sparse initialization approach thus complements any sparsity-encouraging property of the $p$-norm part of the density-diversity penalty.

## 3.3 COMPRESSION WITH LOW DIVERSITY AND HIGH SPARSITY

Low diversity and high sparsity both can significantly reduce the number of bits required to encode the weight matrices of the trained network. Specifically, for low diversity, considering the weight matrix with $d$ distinct entries, we only need $\lceil log_2 d \rceil$ bits to encode each entry, in contrast to 32-bit floating point values for the original, uncompressed model. High sparsity facilitates encoding the weight matrices in a standard, sparse matrix representation using value and position pairs. Therefore, for a weight matrix in which $s$ entries are not equal to the modal value of the matrix, $2s + min(r_j, c_j)$ entries are required for encoding, where the $min(r_j, c_j)$ part is for indicating the row or column index, depending on whether compressed sparse row or column form is used. We note that further compression is possible: e.g., by encoding sparsity with values and increments in positions instead of absolute positions, so that the increments are often small values which require less bits to encode. Huffman Coding can also further be applied in the final step, as is done in (Han et al., 2015a), but we do not report this method in our results.

## 3.4 TYING THE WEIGHTS

Because the density-diversity penalty collapses weight matrix entries into same values, we tie the weights together so that for every distinct value $v$ of weight matrix $W_j$, the entries that are equal to $v$ are updated with the average of their gradients.

In practice, we design the training procedure to alternate between learning with the density-diversity penalty and learning with tied weights. During the phase of learning with the density-diversity penalty, we apply the density-diversity penalty with untied weights. This approach greatly reduces the diversity of the weight matrix entries. However, the performance of the resulting model could be inferior to that of the original model because this approach does not optimize the loss function directly. Therefore, for the phase of learning with tied weights, we train the network without the density-diversity penalty, but we tie the entries of the weight matrices according to the pattern learned from the previous phase. In this way, the network's performance improves while the diversity patterns of the weight matrices are unchanged. We also note that during the phase of learning with tied weights, the sparsity pattern is also fixed, because it was learned in the previous density-diversity pattern learning phase. We show the full procedure of model training with the density-diversity penalty in Figure 1.

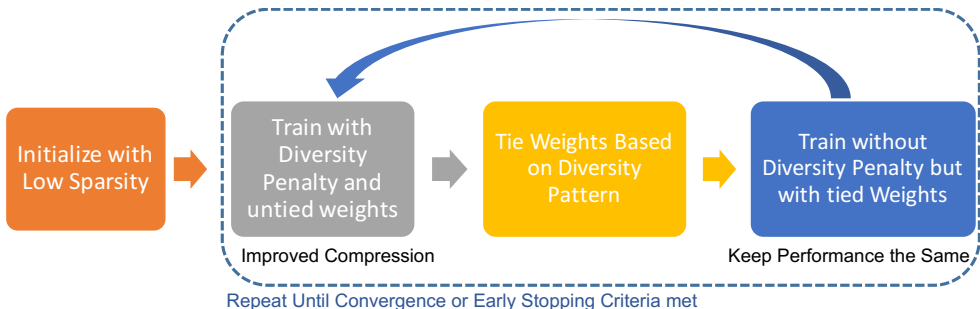

Figure 1: **Pipeline for compressing networks using the density-diversity penalty.** Initialization with low sparsity encourages the weights to collapse, as enforced by the density-diversity penalty. Training with the density-diversity penalty greatly compresses the network by increasing sparsity and decreasing diversity, but the resulting performance can be suboptimal. Training with the tied weights boosts the performance so that we obtain a highly compressed model with the same performance as the original model.

An alternative approach would be to train the model with tied weights and use the density-diversity penalty simultaneously. However, because the diversity pattern, which controls the tying of the weights, changes rapidly across mini-batches, the weights would need to be re-tied frequently based on the latest diversity pattern. This approach would therefore be computationally expensive. Instead, we choose to alternate between applying the density-diversity penalty and training tied weights. In practice, we train each phase for 5 to 10 epochs.

We note that the three key components of our algorithm, namely the sparse initialization, the regularization, and the weight tying, contribute to highly compressed network only when they are applied jointly. Independently applying any of the key component would result in inferior results (we stated developing our approach without the sparse initialization and weight tying and got worse compression results) as a good compression requires low diversity, which is achieved by applying the density-diversity penalty, high sparsity, which is achieved by applying both the density-diversity penalty and the sparse initialization, and no loss of performance, which is achieved by training with weight tying.

## 4  RESULTS

We apply the density-diversity penalty (with $p = 2$ for now) to the fully-connected layers of the models on both the MNIST (computer vision) and TIMIT (speech recognition) datasets, and get significantly sparser and less diverse layer weights. This approach yields dramatic compression of models, whose original sizes are already quite conservative, while keeping the performance unchanged. For our implementation, we start with the mxnet (Chen et al., 2015a) package, which we modified by changing the weight updating code to include our density-diversity penalty.

To evaluate the effectiveness of the density-diversity penalty, we report the diversity and sparsity of the trained weight matrices. We define "diversity" to be number of distinct values divided by the total number of entries in the weight matrix , "sparsity" to be number of entries with the modal value divided by the total number of entries, and "density" to be $1 - $ sparsity. Based on the observed diver-

| Layer | # Weights | DP Density | DC Density | DP Diversity | DC Diversity |
|-------|-----------|------------|------------|--------------|--------------|
| fc1 | 235K | 0.025 | 0.08 | 0.0031 | 0.0003 |
| fc2 | 30K | 0.11 | 0.09 | 0.017 | 0.0021 |
| fc3 | 1K | 0.8 | 0.26 | 0.77 | 0.064 |
| Overall | 266K | 0.037 | 0.08 | 0.018 | 0.0007 |

Table 1: Compression for LeNet-300-100 on MNIST, comparing model trained with density-diversity penalty (DP) and "deep compression" method (DC).

| Layer | # Weights | DP Density | DC Density | DP Diversity | DC Diversity |
|-------|-----------|------------|------------|--------------|--------------|
| conv1 | 0.5K | 1.0 | 0.66 | 1.0 | 0.51 |
| conv2 | 25K | 1.0 | 0.12 | 1.0 | 0.010 |
| fc1 | 400K | 0.0034 | 0.08 | 0.0017 | 0.0002 |
| fc2 | 5K | 0.048 | 0.19 | 0.042 | 0.013 |
| Overall FC | 405K | 0.0039 | 0.08 | 0.0022 | 0.0003 |
| Overall | 431K | 0.063 | 0.08 | 0.061 | 0.0014 |

Table 2: Compression for LeNet-5 on MNIST, comparing model trained with density-diversity penalty (DP) and "deep compression" method (DC). The Overall FC row reports the overall statistics for the fully-connected layers only, where density-diversity penalty is applied.

sity and sparsity, we can estimate the compression rate of the model. For weight matrix $W_j$, suppose $k_j^{value} = \lceil log_2(\text{diversity}(W_j) * r_j * c_j) \rceil$, and $k_j^{index} = \lceil log_2(min(r_j, c_j)) \rceil$, which represent the bits required to encode the value and position, respectively, in the sparse matrix representation. Thus, we have

$$\text{CompressRate}(W_j) = \frac{r_j c_j p}{(1 - \text{sparsity}(W_j)) r_j c_j (k_j^{value} + k_j^{index}) + \text{diversity}(W_j) r_j c_j p + min(r_j, c_j)},$$

(4)

where $p$ is the number of bits required to encode the weight matrix entries used in the original model, and we choose $p = 32$ as used in most modern neural networks.

## 4.1 MNIST DATASET

The MNIST dataset consists of hand-written digits, containing 60000 training data points and 10000 test data points. We further sequester 10000 data points from the training data to be used as the validation set for parameter tuning. Each data point is of size $28 \times 28 = 784$ dimensions, and there are 10 classes of labels.

We choose LeNet (LeCun et al., 1998) as the model to compress, because LeNet performs well on MNIST while having a restricted size, which makes compression hard. Specifically, we test on LeNet-300-100, which consists of two hidden layers, with 300 and 100 hidden units respectively, as well as LeNet-5, which contains two convolutional layers and two fully connected layers. Note that for LeNet-5, we only apply the density-diversity penalty on the fully connected layers. For optimization, we use SGD with momentum.

We report the diversity and sparsity of each layer of the LeNet-300-100 model trained with the diversity penalty in Table 1. The overall compression rate for the LeNet-300-100 model is 32.43X, using 10 bits to encode both value and index of the sparse matrix representation (the number of bits are based on the number of distinct values in the trained weight matrices). The error rate of the compressed model is 1.62%, while the error rate of the original model is 1.64%; thus, we obtain a highly compressed model without loss of performance. Compared to the state-of-the-art "deep compression" result (Han et al., 2015b), which achieves roughly 32 times compression rate (without applying Huffman Coding in the end), our method overall achieves a better compression rate. We also note that "deep compression" uses a more complex sparsity matrix representation, so that indices of values are encoded with many fewer bits.

In Table 2, we show the per-layer diversity and sparsity of the LeNet-5 convolutional model applied with the density-diversity penalty. For such a model, the overall compression rate is 15.78X. When considering only the fully-connected layers of the model, where most parameters reside and the

| Layer | # Weights | DP Density | DC Density | DP Diversity | DC Diversity |
|-------|-----------|------------|------------|--------------|--------------|
| fc1 | 3778K | 0.037 | 0.12 | 0.0004 | 1.6e-5 |
| fc2 | 4194K | 0.064 | 0.13 | 0.0003 | 1.5e-5 |
| fc3 | 4194K | 0.080 | 0.14 | 0.0004 | 1.5e-5 |
| fc4 | 251K | 0.35 | 0.25 | 0.012 | 0.0002 |
| Overall | 12417K | 0.0947 | 0.1936 | 0.0007 | 1.9e-5 |

Table 3: Compression statistics for 3-2048 fully-connected network on TIMIT dataset, comparing model trained with density-diversity penalty (DP) and "deep compression" method (DC).

density-diversity penalty applies, the compression rate is 226.32X, using 9 bits for both value and index for sparse matrix representation. The error rate for the compressed model is 0.93%, which is comparable to the 0.88% error rate of the original model. Compared to the "deep compression" method (without Huffman Coding), which gives 33 times compression for the entire model, and 36 times compression for the fully-connected layers only, our approach achieves better results on the fully-connected layers.

## 4.2 TIMIT DATASET

The TIMIT dataset is for a speech recognition task. The dataset consists of a 462 speaker training set, a 50 speaker validation set, and a 24 speaker test set. Fifteen frames are grouped together as inputs, where each frame contains 40 log mel filterbank coefficients plus energy, along with their first and second temporal derivatives (Mohamed et al., 2012). Overall, there are 1.1M training data samples, 120k validation samples, and 50k test samples. We use a window of size $15 \pm 7$ of each frame, so that each data point has 1845 dimensions. Each dimension is normalized by subtracting the mean and dividing by the standard deviation. The label vector has 183 dimensions, consisting of three states for each of the 61 phonemes. For decoding, we use a bigram language model (Mohamed et al., 2012), and the 61 phonemes are mapped to 39 classes as done in (Lee & Hon, 1989) and as is quite standard.

We choose the model used in (Mohamed et al., 2012) and (Ba & Caruana, 2014) as the target for compression. In particular, the model contains three hidden fully-connected layers, each of which has 2048 hidden units. We choose ReLU as the activation function and AdaGrad (Duchi et al., 2011) for optimization, which performs the best on the original models without the density-diversity penalty.

Table 3 shows the per-layer diversity and sparsity for both the density-diversity penalty and "deep compression" applied to the 3-2048 fully-connected model on TIMIT dataset. We train the original, uncompressed model and observe a 23.30% phone error rate on the core test set. With our best effort of tuning parameters for the "deep compression" method, we get 23.35% phone error rate and a compression rate of 19.47X, using 64 cluster centers for the $k$-means quantization step. For our density-diversity penalty regularization, we get 21.45X compression and 23.25% phone error rate with 11 digits for value encoding and 11 digits for position encoding for the sparse matrix representation.

We visualize a part of the weight matrix either trained with or without the density-diversity penalty in Figure 2. We clearly observe that the weight matrix trained with the density-diversity penalty has significantly more commonality amongst entry values. In addition, the histogram of the entry values comparing the two weight matrices (Figure 3) shows that the weight matrix trained with density-diversity penalty has much less variance in the entry values. Both figures show that density-diversity penalty effectively makes the weight matrix extremely compressed.

## 5 DISCUSSION

On both the MNIST and TIMIT datasets, compared to the "deep compression" method (Han et al., 2015b), the density-diversity penalty achieves comparable or even better compression rates on fully-connected layers. Another advantage offered by the density-diversity penalty approach is that, rather than learning the sparsity pattern and diversity pattern separately as done in the "deep compression" method, the density-diversity penalty enforces high sparsity and low diversity simultaneously, which greatly reduces the effort involved in tuning parameters.

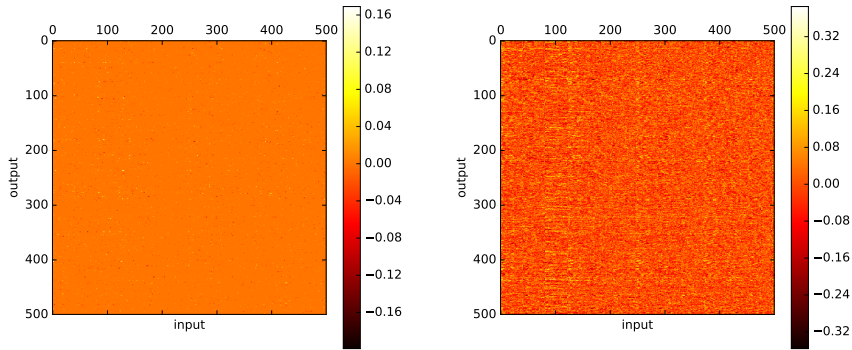

Figure 2: Visualization of the first 500 rows and 500 columns of the first layer weight matrix (shape 2048 X 1845) of the TIMIT 3-2048 model, comparing training either with (left) or without density-diversity penalty (right).

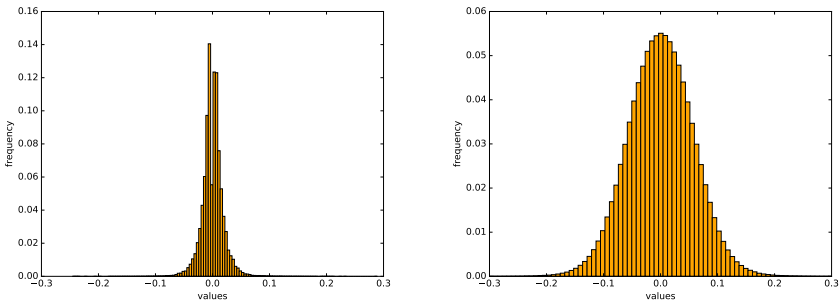

Figure 3: Histogram of the entries of the first layer weight matrix (shape 2048 X 1845) of the TIMIT 3-2048 model with zero entries removed, comparing training either with (left) or without density-diversity penalty (right).

Specifically, comparing the diversity and sparsity of the trained matrices using the two compression methods, we find that the density-diversity penalty achieves higher sparsity but more diversity than the "deep compression" method. The "deep compression" method has two separate phases, where for the first phase, the sparsity pattern is learned by pruning away low value entries, and for the second phase, k-means clustering is applied to quantize the matrix entries into a chosen number of clusters (e.g., 64), thus generating weight matrices with very low diversity. In contrast, the density-diversity penalty acts as a regularizer, enforcing low diversity and high sparsity simultaneously and during training, so that the diversity and sparsity of the trained matrices are more balanced.

# 6   CONCLUSION

In this work, we introduce density-diversity penalty as a regularization on the fully-connected layers of deep neural networks to encourage high sparsity and low diversity pattern in the trained weight matrices. To efficiently optimize the density-diversity penalty, we propose a "sorting trick" to make the density-diversity penalty computationally feasible. On the MNIST and TIMIT datasets, networks trained with the density-diversity penalty achieve 20X to 200X compression rate on fully-connected layers, while keeping the performance comparable to that of the original model.

In future work, we plan to apply the density-diversity penalty to recurrent models, extend the density-diversity penalty to convolutional layers, and test other values of $p$. Moreover, besides pairwise L1 loss for the diversity portion of the density-diversity penalty, we will investigate other forms of regularizations to reduce the diversity of the trained weight matrices (e.g., other forms of structured convex norms). Throughout this work, we have focused on the compression task, but the learned sparsity/diversity pattern of the trained weight matrices is also worth exploring further. For image and speech data, we know that we can use the convolutional structure to improve performance, whereas for other very different forms of data, where we have no prior knowledge about the structure (i.e., patterns of locality), the density-diversity penalty may be applied to discover the underlying hidden pattern of the data and to achieve improved results.

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
