# Peer review of "Training Compressed Fully-Connected Networks with a Density-Diversity Penalty"

_ICLR 2017 — accepted_

[Official Review · AnonReviewer3 · rating 6 · confidence 4 · 16 Dec 2016]

This work introduces a number of techniques to compress fully-connected neural networks while maintaining similar performance, including a density-diversity penalty and associated training algorithm. The core technique of this paper is to explicitly penalize both the overall magnitude of the weights as well as diversity between weights. This approach results in sparse weight matrices comprised of relatively few unique values. Despite introducing a more efficient means of computing the gradient with respect to the diversity penalty, the authors still find it necessary to apply the penalty with some low probability (1-5%) per mini-batch.

The approach achieves impressive compression of fully connected layers with relatively little loss of accuracy. I wonder if the cost of having to sort weights (even for only 1 or 2 out of 100 mini-batches) might make this method intractable for larger networks. Perhaps the sparsity could help remove some of this cost?

I think the biggest fault this paper has is the number of different things going on in the approach that are not well explored independently. Sparse initialization, weight tying, probabilistic application of density-diversity penalty and setting the mode to 0, and alternating schedule between weight tied standard training and diversity penalty training. The authors don't provide enough discussion of the relative importance of these parts. Furthermore, the only quantitative metric shown is the compression rate which is a function of both sparsity and diversity such that they cannot be compared on their own. I would really like to see how each component of the algorithm affects diversity, sparsity, and overall compression. 

A quick verification: Section 3.1 claims the density-diversity penalty is applied with a fixed probability per batch while 3.4 implies structured phases alternating between application of density-diversity and weight tied standard cross entropy. Is this scheme in 3.4 only applying the density-diversity penalty probabilistically when it is in the density-diversity phase?

Preliminary rating:
I think this is an interesting paper but lacks sufficient empirical evaluation of its many components. As a result, the algorithm has the appearance of a collection of tricks that in the end result in good performance without fully explaining why it is effective.

Minor notes:
Please resize equation 4 to fit within the margins (\resizebox{\columnwidth}{!}{ blah } works well in latex for this)

[Official Review · AnonReviewer2 · rating 6 · confidence 2 · 16 Dec 2016]
**Good results**

The paper shows promising results but it is difficult to read and follow. It presents different things closely related and it is difficult to asses the performance of each one. Diversity, sparsity, regularization term, tying weights. Anyway results are good.

[Official Review · AnonReviewer1 · rating 9 · confidence 4 · 22 Dec 2016]
**Very good paper. Extremely easy to read and understand. Exciting ideas. Very good results. a few typos.**

The method proposes to compress the weight matrices of deep networks using a new density-diversity penalty together with a computing trick (sorting weights) to make computation affordable and a strategy of tying weights.

This density-diversity penalty consists of an added cost corresponding to the l2-norm of the weights (density) and the l1-norm of all the pairwise differences in a layer.

Regularly, the most frequent value in the weight matrix is set to zero to encourage sparsity.

As weights collapse to the same values with the diversity penalty, they are tied together and then updated using the averaged gradient.

The training process then alternates between training with 1. the density-diversity penalty and untied weights, and 2. training without this penalty but with tied weights.

The experiments on two datasets (MNIST for vision and TIMIT for speech) shows that the method achieves very good compression rates without loss of performance.


The paper is presented very clearly,  presents very interesting ideas and seems to be state of the art for compression. The approach opens many new avenues of research and the strategy of weight-tying may be of great interest outside of the compression domain to learn regularities in data.

The result tables are a bit confusing unfortunately.

minor issues:

p1
english mistake: “while networks *that* consist of convolutional layers”.

p6-p7
Table 1,2,3 are confusing. Compared to the baseline (DC), your method (DP) seems to perform worse:
 In Table 1 overall, Table 2 overall FC, Table 3 overall, DP is less sparse and more diverse than the DC baseline. This would suggest a worse compression rate for DP and is inconsistent with the text which says they should be similar or better.
I assume the sparsity value is inverted and that you in fact report the number of non-modal values as a fraction of the total.

[Public Comment · (anonymous) · 28 Jan 2017]
**timit**

I didn't read the paper in detail, but for the TIMIT results the authors reported 23.x PER, this is actually quite poor and quite off from SOTA. SOTA is around 16.5 PER and even end-to-end methods (w/o HMMs) can achieve ~17.6% (see Alex Graves CTC/RNN transducer paper and Jan Chorowski's Attention paper).

[Public Comment · (anonymous) · 31 Jan 2017]
**conv parameters**

Did the author try the method on conv layers? How is the result? On resnet-152, fc layers have only 4% parameters. Thank you very much!
Comment: a related work of learning structurally sparse DNN:

[Final Decision · Program Chairs · 06 Feb 2017]
**ICLR committee final decision**

The reviewers unanimously recommended accepting the paper.